# Experimental study on engineering properties of fiber-stabilized carbide-slag-solidified soil

**Zhang Hongzhou**[1,2]*, **Tian Limei**[2], **Wang Shuang**[2], **Qiao Yanhong**[2]

**1** School of Engineering and Technology, China University of Geosciences Beijing, Beijing, China, **2** School of Architecture and Civil Engineering, Langfang Normal University, Langfang, Hebei, China

* zhanghongzhou@lfnu.edu.cn

**Data Availability Statement:** The relevant data for this study are available on Zenodo at DOI: 10.5281/zenodo.6405435 (https://doi.org/10.5281/zenodo.6405435).

## Abstract

Carbide slag has been used to prepare solidified soil to effectively reduce the stacking and disposal of carbide slag and achieve efficient resource utilization. Because of the significant brittleness and low strength of carbide-slag-stabilized soil, fibers were added to carbide-slag-stabilized soil in this experimental study. The effects of fiber length and fiber content on the unconfined compressive and indirect tensile strengths of carbide-slag-stabilized soil were investigated. The concepts of the density of fibers in solidified soil and the number of fibers in a unit volume solidified soil were proposed, and the effects of fiber distribution density on the mechanical properties of the solidified soil were evaluated. The fibers increased the indirect tensile strength of the carbide-slag-solidified soil, which was significantly higher than the unconfined compressive strength of the solidified soil. The fibers had no significant effect on the unconfined compressive and indirect tensile strengths of the 7 d carbide-slag-solidified soil but increased those of the 28 d carbide-slag-solidified soil. The enhancement effect was the most significant when a 0.3% content of 19 mm long fibers was incorporated into the carbide-slag-solidified soil.

## Introduction

The efficient treatment and utilization of carbide slag is an effective approach to solve the environmental pollution problem of the carbide industry and achieve sustainable development. Many researchers have attempted to utilize calcium carbide slag for soil reinforcement [1–3], and the solidified soil has problems, such as low strength and toughness and susceptibility to cracking [4]. After more than four wetting–drying cycles, the compressive strength of the carbide-slag-stabilized soil fails to satisfy the strength requirements of the subgrade [5, 6].

Using fibers to stabilize soil is an effective soil reinforcement technique. Fibers can significantly increase the strength and toughness of soil and solve soil cracking problems [7, 8]. The addition of fibers effectively improves the tensile strain and reduces the cracking of specimens [9]. Artificial and natural fibers significantly improve the residual shear strength and unconfined compressive strength limits of soil [10]. Polypropylene fibers can improve the compressive strength, shear strength, and deformation resistance of cement-clay mix [11–13]. The incorporation of fibers had obvious effect on delaying the cracking of carbide slag stabilized

**Funding:** This study was funded by Science and Technology Support Project of Langfang (Grant Number 2021013167). The funder had important role in study design, data collection and analysis, decision to publish, or preparation of the manuscript.

**Competing interests:** The authors have declared that no competing interests exist.

soil specimens under load. [14]. The reinforcement effect of the fiber depends on the strength of the interface between the reinforcement and soil, and the mechanical interactions between the reinforcement and soil are mainly adhesion and friction [15]. The stability mechanism of fibers in the soil can be summarized as "bending mechanism" and "intercrossing mechanism" [16]. The reinforcement effect of fibers in the soil is anisotropic [17]. When the orientation angle of a fiber is perpendicular to the direction of the strong contact force, the fiber exerts the maximum effect [18]. Fibers can improve the ductility and compressive resistance of reinforced soil and reduce the sensitivity of soil to water [19]. Currently, research on the effects of fiber length and fiber content on the properties of fiber-stabilized carbide-slag-solidified soil has not been conducted extensively.

In this study, fiber-stabilized carbide-slag-solidified silt was adopted as the research object. Based on unconfined compressive strength and indirect tensile strength tests, the effects of fiber length and fiber content on the performance of carbide-slag-solidified silt was investigated, and the optimum fiber length and content were determined. The concepts of the mass density of fibers in the solidified soil (DFS) and the number of fibers in a unit volume solidified soil (NFS) were proposed. Moreover, the effects of fiber distribution density on the mechanical properties of the solidified soil were evaluated.

In this study, the properties of fiber-carbide slag-solidified silt were studied according to fiber length and fiber content. The research findings are expected to serve as a reference for applying fiber-stabilized carbide-slag-solidified soil in road engineering. It can protect the environment, improve the resource utilization efficiency of carbide slag, reduce the project cost, achieve sustainable development, and achieve remarkable economic and social benefits.

## Materials and methods

### Materials

The soil samples used for the tests were collected from the Airport, New Area of Langfang, Hebei Province, China. The study did not involve private or protected land. According to Test Methods of soils for Highway Engineering (JTG 3430–2020) [20], the physical indexes and particle gradation of test soil are determined. The physical indexes are listed in Table 1, and the particle size distribution is shown in Fig 1, where $C_u$ and $C_c$ are 13 and 1.06, respectively. Before the tests, the soil samples were crushed by soil milling and passed through a 2.36 mm square hole sieve. The soil collected under the sieve was used for testing.

Carbide slag was purchased from Dezhou Shihua Chemical Co., Ltd. The raw calcium carbide slag had a high moisture content and a peculiar smell. After drying and dehydration, it was crushed by soil milling and passed through a 0.075 mm square hole sieve. The carbide slag under the sieve was used for the tests (Fig 2). The main components of the calcium carbide slag are listed in Table 2. The carbide slag contained a large amount of calcium oxide similar to that of quick lime and exhibited significant activity (Table 2).

The fibers were purchased from Changsha Ninxiang Building Materials Co., Ltd. Polypropylene fibers with lengths of 6, 12, and 19 mm were used for testing (Fig 3). According to Testing Method for Length of Man-made Staple Fibers(GB/T 14336–2008) [21] and Testing

**Table 1. Physical indexes of soil used for testing.**

| Property | $\omega_L$ | $\omega_P$ | $I_P$ | $G$ | $\sigma_{dmax}$ | $OMC$ |
|----------|-----------|-----------|-------|-----|----------------|-------|
|          | %         | %         | %     |     | g/cm$^3$       | %     |
| Value    | 31.1      | 22.8      | 8.3   | 2.45 | 1.86          | 12.15 |

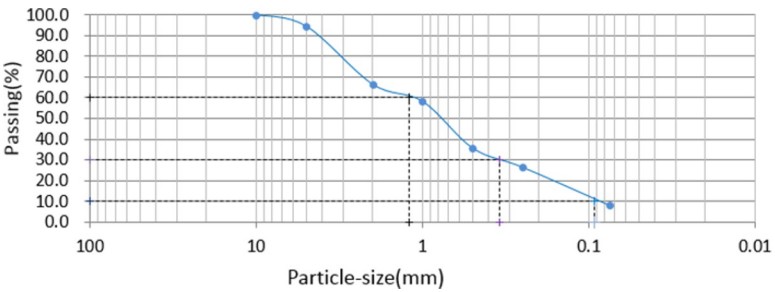

**Fig 1. Particle-size distribution of the soil.** Before the tests, the soil samples were crushed by soil milling and passed through a 2.36 mm square hole sieve. The soil collected under the sieve was used for testing. D60 of the soil is 1.208mm, D30 of the soil is 0.344mm, D10 of the soil is 0.093mm, $C_u$ and $C_c$ of the soil are 13 and 1.06.

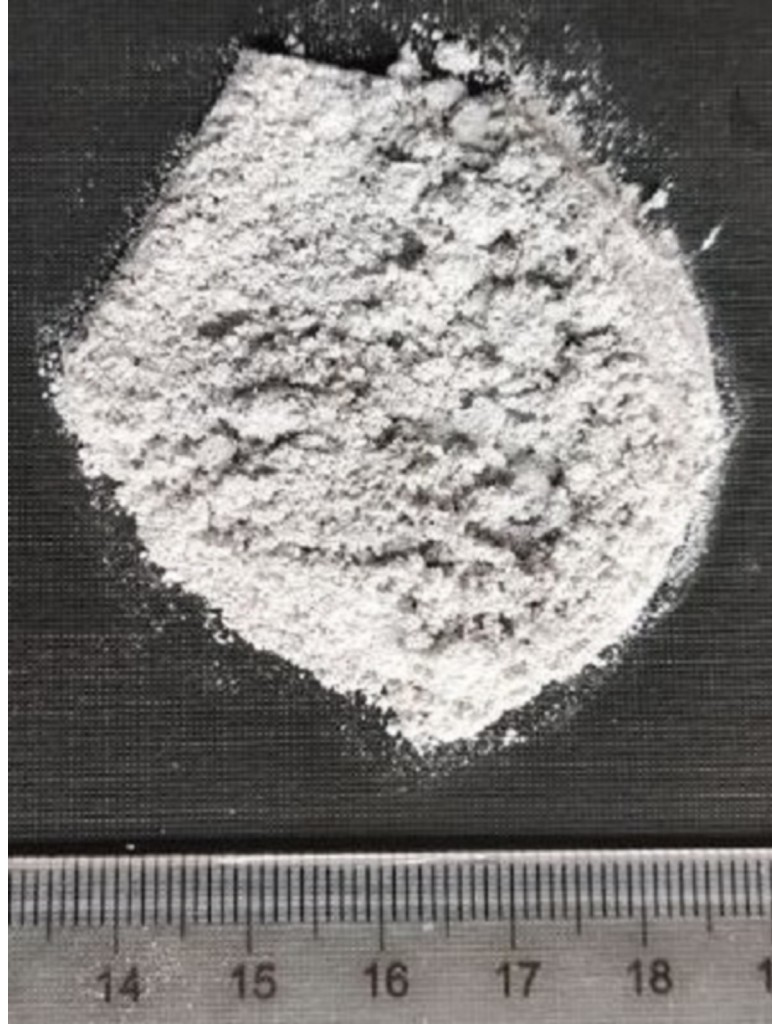

**Fig 2. The carbide slag used in the study.** Because carbide slag contains a large amount of calcium hydroxide, it has strong hygroscopicity, so it needs to be dried before use. After drying and dehydration, it was crushed by soil milling and passed through a 0.075 mm square hole sieve.

**Table 2. Chemical composition of carbide slag.**

| Chemical compound | CaO | SiO$_2$ | Al$_2$O$_3$ | Fe$_2$O$_3$ | MgO | Loss on ignition |
|---|---|---|---|---|---|---|
| Content (%) | 65.77 | 3.15 | 1.78 | 0.24 | 0.14 | 23.56 |

Method for Tensile Properties of Man-made Staple Fibers(GB/T 14337–2008) [22], the physical and mechanical properties of the fibers are listed in Table 3.

## Methods

This study was divided into three steps to investigate the effects of the fiber length and content on the performance of fiber-stabilized carbide-slag-solidified silt, optimal fiber length and content, and effect of fiber distribution density on the mechanical properties of the solidified soil.

1. Compaction tests. The optimum moisture content and the maximum dry density of carbide-slag-solidified silt were determined according to "Compaction Test Method for Inorganic Binder Stabilized Materials" (T0804-1994).

2. Unconfined compressive strength tests. The unconfined compressive strength of fiber-stabilized carbide-slag-solidified silt was determined according to the "Unconfined Compressive Strength Test Method for Inorganic Binder Stabilized Materials" (T0805-1994).

3. Indirect tensile strength test. The indirect tensile strength of fiber-stabilized calcium-carbide-slag-solidified silt was determined according to "Indirect Tensile Strength Test Method for Inorganic Binder Stabilized Materials (Splitting Test)" (JTGE51-2009-T0806).

## Preparation of specimens

The specimens were prepared using the method specified in "Inorganic Binder-Stabilized Material Specimen Production Method (Cylindrical)" (T0843-2009). Dried sieved carbide slag and silt were thoroughly blended at a mass ratio of 2:8. An appropriate amount of the fiber was proportionally added to the mixture and thoroughly stirred manually to fully disperse the fiber into the mixture. An appropriate amount of water was added to the mixture blended with the fiber, stirred evenly, and placed in a plastic bag. The bag was sealed and placed in the dark for 24 h. A 50 mm × 50 mm test mold was adopted, and the calculated weight of the soil sample was determined based on the degree of compaction (96%). Twelve samples were prepared for each group. After the specimens were produced using the static pressure method, they were placed in a curing box (temperature = 20±2˚C; relative humidity ≥ 95%). The specimens are shown in Fig 4. The proportions of the specimens of fiber-stabilized carbide-slag-solidified soil are listed in Table 4.

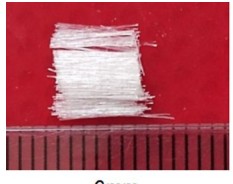
6mm

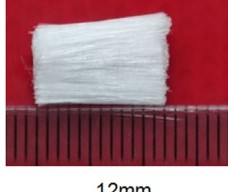
12mm

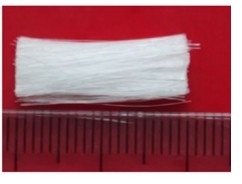
19mm

**Fig 3. The fibers used for the study.** The fibers used for the study is polypropylene monofilament staple fibers, which have high strength and elastic modulus, excellent dispersion, no agglomeration, stable chemical properties, acid and alkali resistance.

**Table 3. Physical and mechanical properties of polypropylene fibers.**

| Length (mm) | Diameter (μm) | Density (g/cm³) | Tensile strength (*MPa*) | *E* (*MPa*) | Ultimate elongation (%) | Melting point (˚C) |
|---|---|---|---|---|---|---|
| 6/12/19 | 32.7 | 0.91 | 469 | 4236 | 28.4 | 169 |

## Results

### Compaction

According to the experiment, the optimum moisture content of 2:8 carbide-slag-solidified soil was 13.5%. First, five portions of 2:8 carbide-slag-solidified soil were prepared, and the target moisture contents were 10%, 12%, 14%, 16%, and 18%. Compaction tests were performed according to the compaction test method for inorganic binder stabilization materials (T0804-1994). The results of compaction test are shown in S1 Table. The compaction curve of the soil samples is shown in (Fig 5). The optimum moisture content of 2:8 carbide-slag-solidified soil was 13.62%, and the maximum dry density was 1.68g/cm³. Because the amount of fiber was small, its effects on the optimum moisture content and maximum dry density were not evident. Therefore, when preparing an unconfined compressive strength test and indirect tensile strength test samples, it could be directly configured according to the optimum water content of 13.62% and the maximum dry density of 1.68g/cm³.

### Unconfined compressive strength

The failure pattern of the specimens subjected to unconfined compression is shown in Fig 6. The unconfined compressive strength test results of 2:8 carbide-slag-solidified soil with different fiber lengths and contents at 7 and 28 d are shown in Fig 7 and S2 Table.

The incorporation of fibers somewhat improved the unconfined compressive strength of the solidified soil, but the improvement was limited. For the same fiber length and content, the unconfined compressive strength of the solidified soil significantly increased with increasing

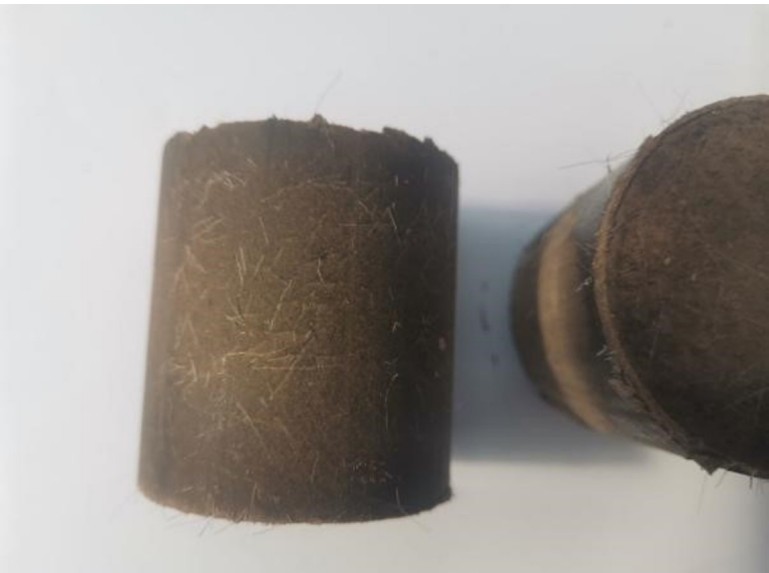

**Fig 4. Prepared specimen of fiber stabilized carbide slag solidified soil.** The size of the specimens were 50 mm in diameter, 50 mm in height, and the degree of compaction was 96%. After the specimens were produced using the static pressure method, they were placed in a curing box (temperature = 20±2˚C; relative humidity ≥ 95%).

**Table 4. Proportions of specimens.**

| Fiber length (mm) | Fiber content (%) | Curing period (d) | Parameter |
|---|---|---|---|
| 6 | 0/0.1/0.2/0.3/0.4 | 7/28 | Unconfined compressive strength test |
| | | | Indirect tensile strength test |
| 12 | 0.1/0.2/0.3/0.4 | 7/28 | Unconfined compressive strength test |
| | | | Indirect tensile strength test |
| 19 | 0.1/0.2/0.3/0.4 | 7/28 | Unconfined compressive strength test |
| | | | Indirect tensile strength test |

curing time. For each group of fibers, the 7 and 28 d unconfined compressive strength increased with increasing fiber content, but the degree of improvement was limited. When the fiber contents were 0.3% and 0.4%, the variation in the unconfined compressive strength was insignificant. For the solidified soil reinforced with a 6 mm fiber at a 0.3% fiber content, the maximum values of the 7 and 28 d unconfined compressive strengths were 1.04 and 1.67 MPa, respectively. For the solidified soil mixed with a 12 mm fiber, the maximum value of the 7 d unconfined compressive strength (1.06 MPa) was observed at the 0.4% fiber content. The maximum unconfined compressive strength at 28 d was 1.69 MPa when the fiber content was 0.3%. For the solidified soil mixed with a 19 mm fiber, the maximum unconfined compressive strengths at 7 and 28 d were 1.06 and 1.72 MPa, respectively, when the fiber content was 0.4%. With an increase in the fiber length, the unconfined compressive strength of the solidified soil increased slightly, but the degree of increase was minimal.

## Indirect tensile strength

The failure patterns of the test specimens subjected to indirect tensile strength test is shown in Fig 8. The indirect tensile strength test results for the calcium-carbide-slag-solidified soil at 7 and 28 d in 2:8 ratio with different fiber lengths and contents are presented in Fig 9 and S3 Table. The addition of fiber did not influence the indirect tensile strength of the solidified soil at 7 d but significantly increased the indirect tensile strength at 28 d (Fig 9). For the same fiber length and content, the indirect tensile strength of the solidified soil significantly increased with increasing curing time. With an increase in the fiber length for the same fiber content, the indirect tensile strength of the 7 d solidified soil did not increase significantly; in contrast, the indirect tensile strength of the 28 d solidified soil increased significantly. When the fiber length was 6 mm, and the fiber content was 0.4%, the maximum indirect tensile strength at 7 d was 0.071 MPa. The maximum indirect tensile strength at 28 d was 0.156 MPa when the fiber content was 0.3%. When the fiber length was 12 mm using a fiber content of 0.3%, the maximum indirect tensile strength at 28 d was 0.156 MPa. The maximum indirect tensile strength of the 7 d cured soil was 0.073 MPa when the fiber content was 0.4%. The maximum indirect tensile strength of the cured soil at 28 d was 0.179 MPa when the fiber content was 0.3%. The maximum indirect tensile strength of the 7 d specimen produced with 19 mm fiber at 0.3% content was 0.075 MPa. When the fiber content was 0.3%, the maximum indirect tensile strength of the 28 d solidified soil was 0.189 MPa.

## Justifications of the results

Because fibers are stable materials, they do not react with silt or calcium carbide slag, and their effect on soil is similar to the mechanism by which plant roots solidify in soil. According to the bending mechanism, the bending fibers between soil particles limit the change in the relative position of the soil particles through the pressure and frictional force on the concave side of

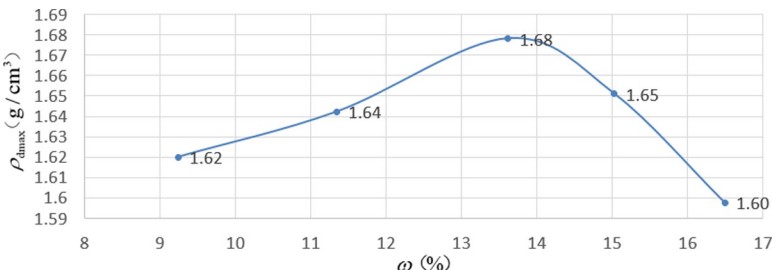

**Fig 5. Compaction curve of carbide slag solidified soil.** According to the experiment, the optimum moisture content of 2:8 carbide-slag-solidified soil was 13.62%, and the maximum dry density was 1.68g/cm³.

the soil particles, stabilizing the soil. According to the intercrossing mechanism, the intercrossing fibers in soil form a spatial network structure, and the action of external forces in one direction is counteracted by the entire network structure. In addition, the incorporation of fibers effectively reduces the internal pores of solidified soil, densifies the soil, and improves its mechanical properties; this can be called the "filling mechanism." Based on the bending, intercrossing, and filling mechanisms, the preconditions for fiber stability in the soil are that the fiber must have a sufficient length and minimum distribution density in the soil.

## Discussion

### Effect of fiber on unconfined compressive strength of solidified soil

Based on the stabilization mechanism of fibers on soil, fibers do not affect the cementing action between the carbide slag and soil. Additionally, the unconfined compressive strength of carbide-slag-solidified soil is mainly influenced by the degree of soil compaction, carbide slag dosage, and soil group [9]. Hence, the fiber had no significant effect on the unconfined compressive strength of the solidified soil. The maximum unconfined compressive strength of the 7 d cured soils were observed when 19 mm fiber was added at a 0.4% content. Compared with the cured soil without fiber, the unconfined compressive strength increased by 8.16%. The fibers filled the pores between the particles of the solidified soil mixture and improved the

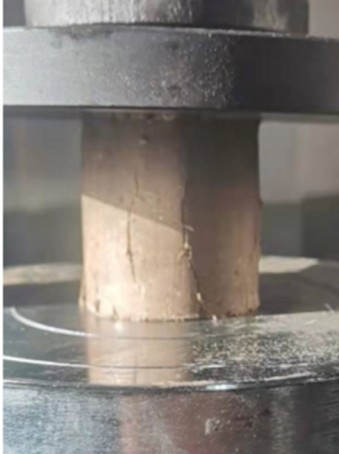
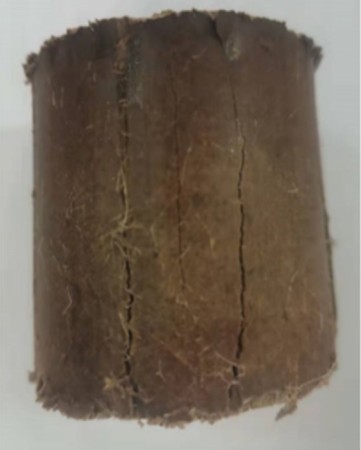

**Fig 6. Failure of the unconfined compressive test specimen.** There are many and small cracks on the surface of the sample. When the failure of the sample is serious, no fracture surface has been formed, resulting in the sliding of the soil block. At the same time, obvious expansion occurs at the end of the sample.

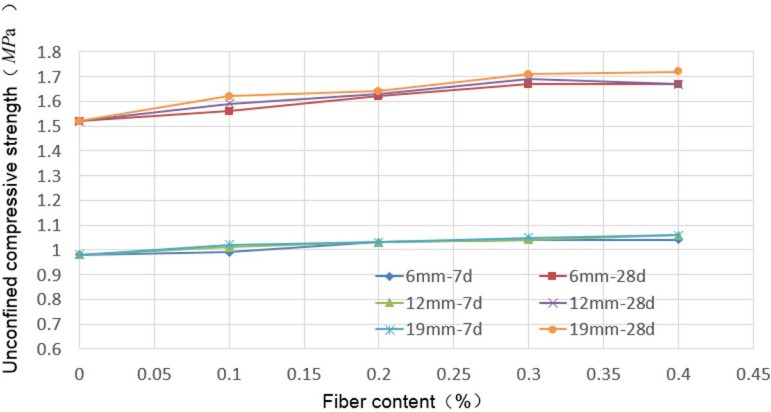

**Fig 7. Unconfined compressive strength of fiber stabilized carbide slag solidified soil.** For the same fiber length and content, the unconfined compressive strength of the solidified soil significantly increased with increasing curing time. For each group of fibers, the 7 and 28 d unconfined compressive strength increased with increasing fiber content, but the degree of improvement was limited.

density of the solidified soil, and reduced the void ratio of the solidified soil. Under the action of external load, the contact area between the fibers and the particles of the solidified soil mixture would generate friction, which played a constraint role on the slip of the particles of the solidified soil mixture. Therefore, the unconfined compressive strength of the 7 d solidified soil reinforced with fibers increased. Its increasing value was related to fiber incorporation: When the content of fiber was higher, the contact area between the fibers and the particles of the solidified soil mixture was greater, the friction would be more evident, and the enhancement of the unconfined compressive strength would be more evident. The maximum unconfined compressive strength of the 28 d solidified soil was observed when a 0.4% content of the 19 mm fiber was added. Compared with the solidified soil without fiber, the unconfined compressive strength increased by 13.16%. With increasing curing time, cementation reaction between the carbide slag and soil particles occurred, and the newly generated reaction products adhered to the spatial network structure formed by the fibers, limiting soil deformation and increasing the unconfined compressive strength of the soil. For the same carbide slag

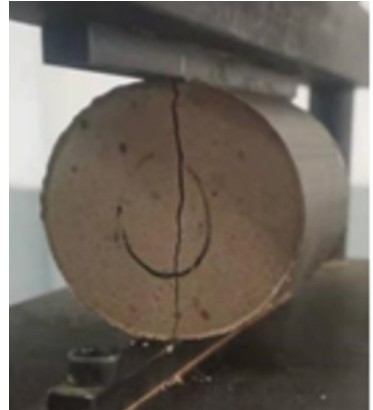
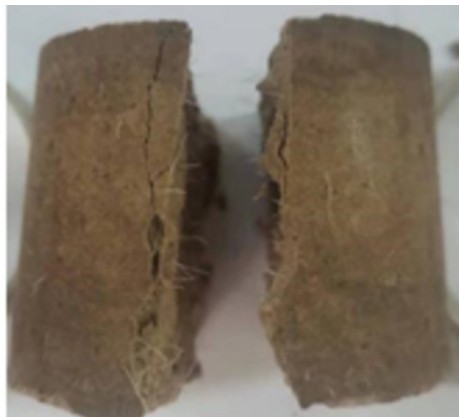

**Fig 8. Indirect tensile strength test specimen.** Under the action of static load, the failure of specimen shows three processes: initial crack generation, crack propagation, new crack generation and fiber pulling out. After cracking, crack development and fiber pulling out will be carried out simultaneously.

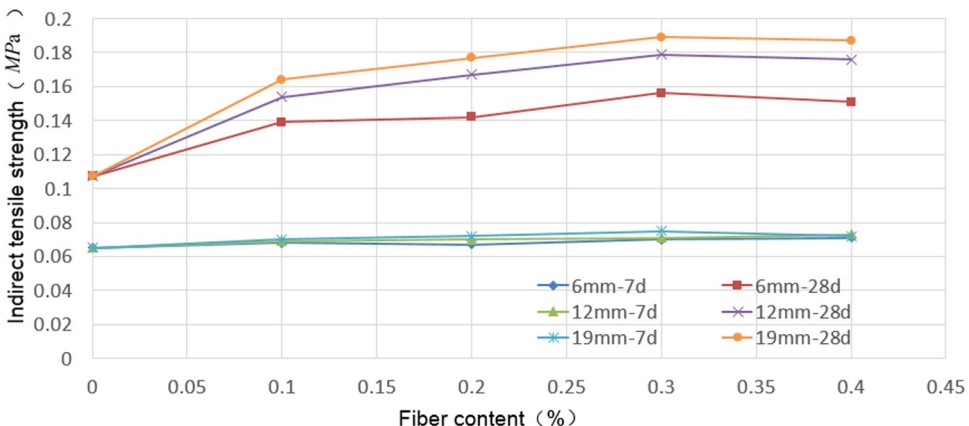

**Fig 9. Results of indirect tensile strength test.** For the same fiber length and content, the indirect tensile strength of the solidified soil significantly increased with increasing curing time. With an increase in the fiber length for the same fiber content, the indirect tensile strength of the 7 d solidified soil did not increase significantly; in contrast, the indirect tensile strength of the 28 d solidified soil increased significantly.

content and soil quality, the intercrossing mechanism effect became more significant with increasing fiber content and length. Moreover, the spatial network structure stabilized further, and the increase in the unconfined compressive strength of the solidified soil became significant. However, when the fiber content exceeded a specific value, the fibers were less uniformly distributed in the soil, and the spatial network structure became less stable.

## Effect of fiber on indirect tensile strength of solidified soil

Based on the stability mechanism of the fiber to the soil, the fiber did not affect the cementation between the carbide slag and soil. The effect of the fiber on the indirect tensile strength of the carbide-slag-solidified soil was mainly achieved by bending and intercrossing. For the 7 d solidified soil, the cementing action between the carbide slag and soil particles was incomplete, and an effective bond between the fiber and solidified soil was not established. Therefore, the effect of the fiber on the indirect tensile strength of the 7 d solidified soil was insignificant. The maximum indirect tensile strength of the 7 d solidified soil was observed for the fiber length of 19 mm (Fig 9). When the added amount was 0.3%, the indirect tensile strength of the solidified soil increased by 15.38% compared with that of the solidified soil without fiber. However, when the fiber content was 0.4%, the indirect tensile strength of the solidified soil decreased. This trend occurred because when the fiber length was extended, a high quantity of the fiber in the stabilized soil was difficult to distribute uniformly, easily formed clusters, reduced the bending effect, and intertwined. For the 28 d solidified soil, cementing between the carbide slag and soil particles was relatively complete, and adequate bonding between the fiber and solidified soil was achieved. Therefore, the fiber significantly increased the indirect tensile strength of the 28 d solidified soil. The maximum indirect tensile strength of the 28 d solidified soil was observed using 19 mm long fibers. When the added amount was 0.3%, the indirect tensile strength of the solidified soil was 76.64% higher than that of the solidified soil without fibers. However, when the fiber content was increased to 0.4%, the indirect tensile strength of the solidified soil decreased, mainly because of difficulty in achieving uniform distribution with a relatively high amount of extended fibers; this easily induced the cluster phenomenon, reduced the bending effect, and caused intertwining.

## Effect of fiber distribution density on mechanical properties of solidified soil

In this study, three fiber lengths (6, 12, and 19 mm) were adopted, and the incorporation amounts were 0.1%, 0.2%, 0.3%, and 0.4% of the total mass of solidified soil. The mass density of fibers in the solidified soil is denoted $\sigma_f$, and the fibers of lengths 6, 12, 19 mm are denoted $\sigma_{f6}$, $\sigma_{f12}$, and $\sigma_{f19}$, respectively. The number of fibers in the solidified soil per unit volume is denoted $N_f$, and the number of fibers of lengths 6, 12, and 19 mm are denoted $N_{f6}$, $N_{f12}$, and $N_{f19}$, respectively. In this study, the diameters for the three fiber lengths (6, 12, and 19 mm) were consistent when the $\sigma_f$ of the specimens were the same: $N_{f19} = 1.5N_{f12} = 3N_{f6}$; that is, for an equal mass, the shorter the fiber, the more the number of fibers in the unit volume of the solidified soil. When the $N_f$ values of the specimens were the same, $\sigma_{f19} = 1.5\sigma_{f12} = 3\sigma_{f6}$; that is, in the case of an equal number of fibers in the unit-volume solidified soil, the longer the fiber, the greater the mass density of fiber in the solidified soil of unit volume.

When the mass density of the 6 mm fiber in the solidified soil was $\sigma_{f6} = 0.1\%$, the mass density of fiber in the solidified soil reinforced with 12 and 19 mm fibers were $\sigma_{f12} = 0.2\%$ and $\sigma_{f19} = 0.3\%$, respectively. The results of the unconfined compressive test obtained at 7 and 28 d are presented in Fig 10, and those of the indirect tensile test are presented in Fig 11.

Under the same conditions, the unconfined compressive strength of the 7 d solidified soil slightly increased with the fiber content (Fig 10). The unconfined compressive strength of the solidified soil reinforced with 19 mm fibers was the highest but only 7.14% higher than that without fibers. The main reason for this was that for the short curing period, the fibers did not form an effective bond with the solidified soil, and the bending and intercrossing actions were somewhat challenging to achieve. The increased unconfined compressive strength was mainly attributed to the effective reduction of the pores in the solidified soil by the incorporated fibers and the resulting filling effect, densifying the solidified soil. The solidified soil reinforced with 19 mm fibers using $\sigma_{f19} = 0.3\%$ had the highest unconfined compressive strength. The unconfined compressive strength of each solidified soil increased with the fiber content. The unconfined compressive strength of the solidified soil reinforced with 19 mm long fibers was the largest and was 12.5% higher than that of the solidified soil without fibers. This behavior was mainly attributed to the adequate bonding between the fiber and 28 d solidified soil and reflected the bending and

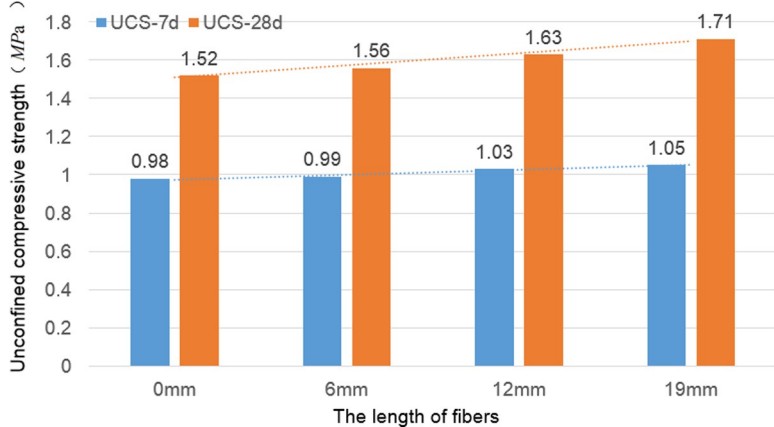

**Fig 10. Results of unconfined compressive test with same $N_f$.** Under the same conditions, the unconfined compressive strength of the solidified soil slightly increased with the fiber content. The unconfined compressive strength of the solidified soil reinforced with 19 mm fibers was the highest.

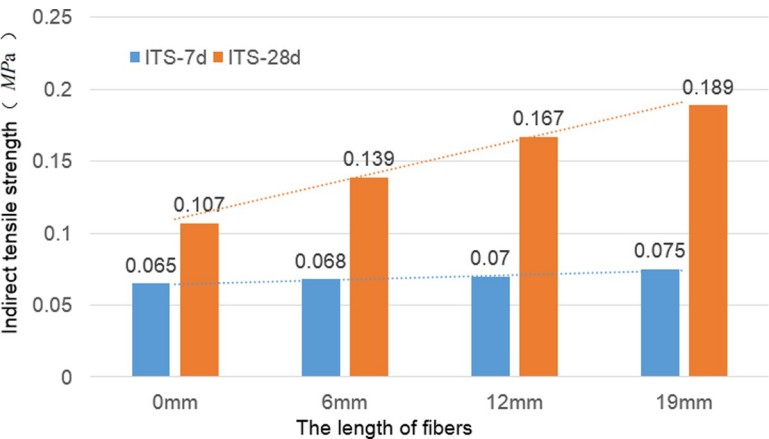

**Fig 11. Results of indirect tensile test with same $N_f$.** Under the same $N_f$ conditions, the indirect tensile strength of the 7d solidified soil increased with the fiber content. The indirect tensile strength of the solidified soil containing 19 mm long fibers was the highest. The 28d indirect tensile strength of each solidified soil increased significantly with the incorporation of fibers. The solidified soil containing 19 mm long fiber exhibited the highest indirect tensile strength.

intercrossing actions. The 19 mm fibers in the solidified soil were the longest, and the content, $\sigma_{f19} = 0.3\%$, was the largest. Moreover, the unconfined compressive strength was the highest under the combined actions of filling, bending, and intercrossing.

Under the same $N_f$ conditions, the indirect tensile strength of the 7 d solidified soil increased with the fiber content (Fig 11). The indirect tensile strength of the solidified soil containing 19 mm long fibers was the highest; this strength was 15.38% higher than that of the solidified soil without fibers. The 28 d indirect tensile strength of each solidified soil increased significantly with the incorporation of fibers. The solidified soil containing 19 mm long fiber exhibited the highest indirect tensile strength, which was 24.34% higher than that of the solidified soil without fibers. The results indicated that the fiber and solidified soil formed an effective bond with extended curing time, and the bending and intercrossing actions were significant.

A comparison between Figs 10 and 11 revealed that the increase in the indirect tensile strength of the fiber-reinforced solidified soil was significantly higher than that in the unconfined compression strength.

## Conclusions

In this study, the effects of fiber length and fiber content on the properties of carbide-slag-solidified soil was investigated based on unconfined compressive strength and indirect tensile strength tests. The following conclusions were drawn.

1. The fibers did not significantly increase the unconfined compressive and indirect tensile strengths of the carbide-slag-solidified soil at 7 d but effectively increased those at 28 d. When 19 mm long fibers were incorporated into the carbide-slag-solidified soil, the effect became most significant for the fiber content of 0.3%.

2. The increase in the indirect tensile strength of the carbide-slag-solidified soil by the fibers was significantly higher than that in the unconfined compressive strength.

3. For short-period (7 d) curing, the stabilization of the carbide-slag-solidified soil by the fibers mainly depended on the filling action. For long-age (28 d) curing, the stabilization by the fibers was caused by the combined actions of filling, intercrossing and bending.

## Supporting information

**S1 Table. Results of compaction test.**
(PDF)

**S2 Table. Results of unconfined compression strength test.**
(PDF)

**S3 Table. Results of indirect tensile strength test.**
(PDF)

## Author Contributions

**Conceptualization:** Zhang Hongzhou.

**Data curation:** Tian Limei, Wang Shuang.

**Formal analysis:** Tian Limei, Wang Shuang, Qiao Yanhong.

**Writing – original draft:** Zhang Hongzhou, Qiao Yanhong.

**Writing – review & editing:** Zhang Hongzhou.

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
