## [Decision Letter · Decision Letter 0]

2 Feb 2022

PONE-D-22-00217Experimental study on engineering properties of fiber-stabilized carbide-slag-solidified soilPLOS ONE

Dear Dr. hongzhou,

Thank you for submitting your manuscript to PLOS ONE. After careful consideration, we feel that it has merit but does not fully meet PLOS ONE’s publication criteria as it currently stands. Therefore, we invite you to submit a revised version of the manuscript that addresses the points raised during the review process.

We look forward to receiving your revised manuscript.

Kind regards,

Ahmed Salih Mohammed

Academic Editor

PLOS ONE

Journal Requirements:

"This study was funded by Science and Technology Support Project of Langfang(Grant Number 2021013167)，Science and Technology Support Project of Langfang Normal University(Grant Number XBQ202113)."

Reviewers' comments:

Reviewer's Responses to Questions

**Comments to the Author**

1. Is the manuscript technically sound, and do the data support the conclusions?

Reviewer #1: Yes

Reviewer #2: Yes

2. Has the statistical analysis been performed appropriately and rigorously? 

Reviewer #1: N/A

Reviewer #2: Yes

3. Have the authors made all data underlying the findings in their manuscript fully available?

Reviewer #1: Yes

Reviewer #2: Yes

4. Is the manuscript presented in an intelligible fashion and written in standard English?

Reviewer #1: Yes

Reviewer #2: Yes

5. Review Comments to the Author

Reviewer #1: Authors should address the following for further considerations:

- The novelty and importance of the work should be stressed.

- Table 1 and Fig. 1, standards used in these tests should be added and properly referenced.

- Table 1: there are errors in the titles LL, PL…etc.

- Standards used in testing of t he fiber (Table 3) should be added.

- Quality of figures should be amended.

- Style of Figures 5, 7, 9, 10 and 11. The titles of the axes should be centered, legends of the figures should be enhanced, and numbers should be more visible.

- Justifications of the results should be added. The readers are more interested in the probable reasons of the expected behavior than only reporting the results.

Reviewer #2: Reviewer

Manuscript Number: PONE-D-22-00217

Title: Experimental study on engineering properties of fiber-stabilized carbide-slag-solidified soil

This study aims to investigate the effect of fiber length and fiber content on the properties of carbide-slag-solidified soil based on unconfined compressive strength and indirect tensile strength tests.

The text is written clearly. The adopted methodology is appropriate. Some issues should be improved to make the article's to be clearer.

However, there are some comments for this study that need to be considered and implemented prior to the publication of this manuscript as follows. There are other comments in the PDF format of the manuscript.

1. Page 1 line 19, in abstract "The concepts of mass density "the density is mass per unit volume the expression "mass density" is an incorrect expression remove the mass

2. Page 1 line 19, in abstract "the root number of fibers " explain the root number? Is it just number of fibers?

3. All "Unconfined Compression Strength" change to 'unconfined compressive strength'

4. On Page #2 "Keywords: "Unconfined Compression Strength" change to " Unconfined Compressive Strength";

5. On Page #2 "Keywords: "mass density of fiber in solidified soil"; is a sentence not a keyword, change to " solidified soil ";

6. On Page #2 "Keywords: "root number of fiber in unit volume solidified soil", is a sentence not a keyword, it is suggested to be removed.

7. Page 3 line 50, "improve the compaction strength" change to " improve the compressive strength"

8. Page 3 line 51, " of cemented clay" is not clear is it clay stabilized with cement? Explain and correct it

9. Page 3 line 52, " the load cracking " is not an appropriate expression, correct it

10. Page 3 line 57, " interlacing mechanism " is not clear for me, explain that, several expressions used in the paper " interlacing, interweaving, interleaving, … what are the differences between them?

11. Page 6 line 111, " The optimal moisture content and the maximum dry density of 2:8 carbide-slag-solidified silt " change to, " The optimum moisture content and the maximum dry density of carbide-slag-solidified silt "

12. Page 8 lines 145 – 146, check that " According to the experience, the optimal moisture content of 2:8 carbide-slag-solidified soil was 13%-14%. " There is only one optimum moisture content or it is the range of optimum moisture content? Change " experience" to "experiment"

13. Change all "optimal moisture content" to "optimum moisture content"

14. Page 8 line 153, "the effects on the optimal moisture content " change to "its effect on the optimum moisture content "

15. Page 8 line 154, " when making" change to " when preparing"

16. Page 9 line 167, " increasing age" change to " increasing curing time"

17. Page 10 line 192, " age" change to " curing time"

18. Page 11 line 218, "compacts the soil" change to " Densify the soil "

19. Page 12 line 227, " affect the gelatin between" gelatin is not an appropriate meaning

20. Page 12 lines 229-230, " soil quality" what is the meaning of it?

21. Page 12 line 238-240, " the higher the fiber content, the more significant the compactness of the solidified soil. Moreover, the unconfined compressive strength was relatively high." the interpretation is unconvincing

22. Page 12 lines 289 to 290

The symbols of fiber intensity did not appear in the PDF format

23. Page 16 lines 312, 322, 325 and 335, use more appropriate expression instead of like interweaving instead of " interleaving effects"

24. Page 17 lines 336, " suggests that" change to " revealed that"

25. For figures, the axes title should be in the middle of the axes

6. PLOS authors have the option to publish the peer review history of their article (what does this mean?). If published, this will include your full peer review and any attached files.

Reviewer #1: **Yes: **Saif Alzabeebee

Reviewer #2: **Yes: **Yousif Ismael Mawlood

---

## [Author Response · Author response to Decision Letter 0]

14 Feb 2022

Dear Reviewers:

We tried our best to improve the manuscript and made some changes in the manuscript. These changes will not influence the content and framework of the paper. And here we did not list the changes but marked in red in revised paper.

We appreciate for Reviewers' warm work earnestly, and hope that the correction will meet with approval. Once again, thank you very much for your comments and suggestions.

Thank you and best regards.

Yours sincerely,

Hongzhou Zhang.

---

## [Decision Letter · Decision Letter 1]

1 Mar 2022

PONE-D-22-00217R1Experimental study on engineering properties of fiber-stabilized carbide-slag-solidified soilPLOS ONE

Dear Dr. hongzhou,

Thank you for submitting your manuscript to PLOS ONE. After careful consideration, we feel that it has merit but does not fully meet PLOS ONE’s publication criteria as it currently stands. Therefore, we invite you to submit a revised version of the manuscript that addresses the points raised during the review process.

We look forward to receiving your revised manuscript.

Kind regards,

Ahmed Salih Ahmed

Academic Editor

PLOS ONE

Reviewers' comments:

Reviewer's Responses to Questions

**Comments to the Author**

1. If the authors have adequately addressed your comments raised in a previous round of review and you feel that this manuscript is now acceptable for publication, you may indicate that here to bypass the “Comments to the Author” section, enter your conflict of interest statement in the “Confidential to Editor” section, and submit your "Accept" recommendation.

Reviewer #1: (No Response)

Reviewer #2: All comments have been addressed

2. Is the manuscript technically sound, and do the data support the conclusions?

Reviewer #1: No

Reviewer #2: Yes

3. Has the statistical analysis been performed appropriately and rigorously? 

Reviewer #1: N/A

Reviewer #2: Yes

4. Have the authors made all data underlying the findings in their manuscript fully available?

Reviewer #1: Yes

Reviewer #2: Yes

5. Is the manuscript presented in an intelligible fashion and written in standard English?

Reviewer #1: Yes

Reviewer #2: Yes

6. Review Comments to the Author

Reviewer #1: Authors should highlight the changes so that the reviewer could see what has been added. We all have limited times and it is difficult to understand all the changes without a highlight. Also, it is advised to include the changes within the reply section to avoid delay in review.

Reviewer #2: Manuscript Number: PONE-D-22-00217 R1

Title: Experimental study on engineering properties of fiber-stabilized carbide-slag-solidified

soil

This study aims to investigate the effect of fiber length and fiber content on the properties of carbide-slag-solidified soil based on unconfined compressive strength and indirect tensile strength tests.

The text is written clearly. The adopted methodology is appropriate. Some issues should be improved to make the article's to be clearer, the manuscript was highly improved, just some correction highlighted in the PDF file

7. PLOS authors have the option to publish the peer review history of their article (what does this mean?). If published, this will include your full peer review and any attached files.

Reviewer #1: No

Reviewer #2: **Yes: **Yousif Ismael Mawlood

---

## [Author Response · Author response to Decision Letter 1]

18 Mar 2022

Thank you for your letter and for the reviewers’ comments concerning our manuscript entitled “Experimental study on engineering properties of fiber-stabilized carbide-slag-solidified soil” (ID: PONE-D-22-00217R1). Those comments are all valuable and very helpful for revising and improving our paper, as well as the important guiding significance to our researches. We have studied comments carefully and have made correction which we hope meet with approval. Revised portion are marked in red in the paper. 

We are deeply sorry for the mistake we made in the first revision! In this time, a marked-up copy of our manuscript that highlights changes made to the original version had been upload that as a separate file labeled 'Revised Manuscript with Track Changes'(from page 38 to 57 of PONE-D-22-00217_R2). At the same time, an unmarked version of our revised paper without tracked changes had been upload this as a separate file labeled 'Manuscript'(from page 9 to 28 of PONE-D-22-00217_R2).And the changes were also within the reply section.Special thanks to you for your good comments.

---

## [Decision Letter · Decision Letter 2]

28 Mar 2022

Experimental study on engineering properties of fiber-stabilized carbide-slag-solidified soil

PONE-D-22-00217R2

Dear Dr. hongzhou,

We’re pleased to inform you that your manuscript has been judged scientifically suitable for publication and will be formally accepted for publication once it meets all outstanding technical requirements.

Kind regards,

Ahmed Mohammed

Academic Editor

PLOS ONE

Additional Editor Comments (optional):

Reviewers' comments:

Reviewer's Responses to Questions

**Comments to the Author**

1. If the authors have adequately addressed your comments raised in a previous round of review and you feel that this manuscript is now acceptable for publication, you may indicate that here to bypass the “Comments to the Author” section, enter your conflict of interest statement in the “Confidential to Editor” section, and submit your "Accept" recommendation.

Reviewer #1: All comments have been addressed

Reviewer #2: All comments have been addressed

2. Is the manuscript technically sound, and do the data support the conclusions?

Reviewer #1: Yes

Reviewer #2: Yes

3. Has the statistical analysis been performed appropriately and rigorously? 

Reviewer #1: Yes

Reviewer #2: Yes

4. Have the authors made all data underlying the findings in their manuscript fully available?

Reviewer #1: Yes

Reviewer #2: Yes

5. Is the manuscript presented in an intelligible fashion and written in standard English?

Reviewer #1: Yes

Reviewer #2: Yes

6. Review Comments to the Author

Reviewer #1: Dear Editor,

Comments of the reviewer have been addressed and the paper can be accepted for publication.

Regards

Dr. Saif Alzabeebee

Reviewer #2: All comments have been addressed by the author and the manuscript possible to be published in the POLOS ONE journal

7. PLOS authors have the option to publish the peer review history of their article (what does this mean?). If published, this will include your full peer review and any attached files.

Reviewer #1: **Yes: **Saif Alzabeebee

Reviewer #2: **Yes: **Yousif Ismael Mawlood

---

## [Editor Report · Acceptance letter]

6 Apr 2022

PONE-D-22-00217R2 

Experimental study on engineering properties of fiber-stabilized carbide-slag-solidified soil 

Dear Dr. Hongzhou:

I'm pleased to inform you that your manuscript has been deemed suitable for publication in PLOS ONE. Congratulations! Your manuscript is now with our production department. 

Kind regards, 

on behalf of

Dr. Ahmed Mohammed 

Academic Editor

PLOS ONE